# Construction and Characterization of Severe Fever with Thrombocytopenia Syndrome Virus with a Fluorescent Reporter for Antiviral Drug Screening

**DOI:** 10.3390/v15051147

**Published:** 2023-05-10

**Authors:** Xiao Wang, Mingyue Xu, Huanhuan Ke, Longda Ma, Liushuai Li, Jiang Li, Fei Deng, Manli Wang, Zhihong Hu, Jia Liu

**Affiliations:** 1Division of Life Science and Medicine, University of Science and Technology of China, Hefei 230027, China; 2State Key Laboratory of Virology, Wuhan Institute of Virology, Center for Biosafety Mega-Science, Chinese Academy of Sciences, Wuhan 430071, China; 3Department of Forensic Medicine, Tongji Medical College of Huazhong University of Science and Technology, Wuhan 430030, China; 4Hubei Jiangxia Laboratory, Wuhan 430200, China

**Keywords:** severe fever with thrombocytopenia syndrome virus, fluorescent virus, NSs deletion, attenuated virulence, high-content antiviral drugs screening

## Abstract

Severe fever with thrombocytopenia syndrome (SFTS) caused by a novel bunyavirus (SFTSV) is an emerging infectious disease with up to 30% case fatality. Currently, there are no specific antiviral drugs or vaccines for SFTS. Here, we constructed a reporter SFTSV in which the virulent factor nonstructural protein (NSs) was replaced by eGFP for drug screening. First, we developed a reverse genetics system based on the SFTSV HBMC5 strain. Then, the reporter virus SFTSV-delNSs-eGFP was constructed, rescued, and characterized in vitro. SFTSV-delNSs-eGFP showed similar growth kinetics with the wild-type virus in Vero cells. We further detected the antiviral efficacy of favipiravir and chloroquine against wild-type and recombinant SFTSV by the quantification of viral RNA, and compared the results with that of fluorescent assay using high-content screening. The results showed that SFTSV-delNSs-eGFP could be used as a reporter virus for antiviral drug screening in vitro. In addition, we analyzed the pathogenesis of SFTSV-delNSs-eGFP in interferon receptor-deficient (IFNAR^−/−^) C57BL/6J mice and found that unlike the fatal infection of the wild-type virus, no obvious pathological change or viral replication were observed in SFTSV-delNSs-eGFP-infected mice. Taken together, the green fluorescence and attenuated pathogenicity make SFTSV-delNSs-eGFP a potent tool for the future high-throughput screening of antiviral drugs.

## 1. Introduction

Severe fever with thrombocytopenia syndrome (SFTS) is an emerging hemorrhagic fever caused by the novel tick-borne bunyavirus SFTSV (recently named Dabie bandavirus) [1]. SFTS is characterized by high fever, thrombocytopenia, potentially lethal hemorrhagic manifestations, multiple organ failure, and death, with a fatality rate of 5–30% in East Asia [2,3,4]. Since the first report of SFTS in China in 2010 [2], it has been reported in Japan, Republic of Korea, Myanmar, Vietnam, Thailand, and Pakistan [5]. The main reservoir and vector of SFTSV is the *Haemaphysalis longicornis* tick, which is widely distributed in the Australasian and Western Pacific regions and has recently expanded to the United States [6,7]. To date, there are no specific antiviral drugs or vaccines against SFTS, and the World Health Organization (WHO) placed the disease on its list of the highest research priorities in 2017 [8,9].

SFTSV is a single-stranded negative-sense enveloped RNA virus. Its genome comprises three segments: large (L), medium (M), and small (S) [2]. The L segment encodes an RNA-dependent RNA polymerase (RdRp) that is responsible for viral RNA replication and transcription. The M segment encodes the glycoproteins Gn and Gc, which mediate viral entry, and the S segment uses an ambisense coding strategy to express the nucleocapsid protein (NP) and non-structural proteins (NSs). NP encapsulates viral genomic RNA and forms ribonucleoprotein complexes with RdRp, which are essential for genome transcription and replication [10,11]. Numerous studies have revealed that SFTSV NSs function as interferon (IFN) antagonists [12,13]. NSs forms inclusion bodies in SFTV-infected cells and sequesters and spatially isolates several key antiviral innate immune molecules into the inclusion bodies. The entrapped molecules are mainly involved in host type I IFN induction and signaling pathways, such as those involving the protein kinase TBK1, RIG-I, interferon regulatory factor (IRF)-3, IRF-7, STAT1, and STAT2 [13,14,15,16,17]. Furthermore, in vivo studies indicated that recombinant SFTSV that lacked NSs had limited pathogenicity in aged ferrets and type I IFN receptor-deficient (IFNAR^−/−^) mice [18,19]. Therefore, NSs are the virulence factors of SFTSV.

Ribavirin, favipiravir (T-705), and calcium channel blockers have been suggested as antiviral candidates for SFTS therapy [20,21,22,23]. However, animal experiments and retrospective clinical studies have revealed that ribavirin has limited antiviral efficacy [4,22]. Although favipiravir has shown more potent efficacy than ribavirin in vitro and in animal models, its efficacy in patients with SFTS has yet to be confirmed [23]. Nifedipine, a calcium channel blocker, is promising because it has shown significant antiviral efficacy in vitro and in vivo, and retrospective clinical data have indicated that it decreased the case fatality rate by >5-fold [21]. Other drugs, including chloroquine (CQ), hexachlorophene, 2′-fluoro-2′-deoxycytidine, amodiaquine, anidulafungin, and IFNs, also showed antiviral efficiency against SFTSV in vitro or in vivo [20,24,25]. However, there is still no approved effective treatment for SFTS, and the development of a specific and effective treatment is urgently needed.

High-content screening (HCS), which allows for automated image acquisition and analysis of fluorescence signals, has also yielded tangible benefits in primary drug discovery and target identification (e.g., RNAi screening) [26]. The throughput capacity of modern HCS platforms has enabled full-scale high-throughput screening (screens ranging from 100,000 to more than 1 million compounds) using image-based endpoints [27]. Recombinant fluorescent reporter viruses can achieve high-throughput screening and provide the real-time activity of tested compounds in large drug libraries using HCS. In this study, we aimed to design and evaluate an SFTSV fluorescent reporter with potential use in high-throughput screening for anti-SFTS treatments. We established a reverse genetics system based on the SFTSV HBMC5 strain and constructed a reporter virus, SFTSV-delNSs-eGFP, with the deletion of the virulence factor NSs. The infectivity and pathogenicity of the reporter virus were compared to those of its parental virus in vitro and in vivo, and its potential for antiviral screening was further analyzed.

## 2. Materials and Methods

### 2.1. Cells and Virus

BSR-T7/5 cells, a derivative of golden hamster kidney cells stably expressing T7 RNA polymerase, were kindly provided by Dr. Lei-Ke Zhang from the Wuhan Institute of Virology, Chinese Academy of Sciences [21] and used for reverse genetics. The cells were cultured in Dulbecco’s modified Eagle’s medium (DMEM, Gibco, Billings, MT, USA) supplemented with 10% fetal bovine serum (FBS, Gibco) and 1 mg/mL of G418 antibiotic (Solarbio, Beijing, China). African green monkey kidney cells (Vero) were grown in DMEM supplemented with 10% FBS. HeLa cells were cultured in Eagle’s minimal essential medium (EMEM, Gibco) supplemented with 10% FBS. All 3 cell lines were cultured at 37 °C with 5% CO_2_.

The SFTSV-HBMC5 strain was isolated from a patient with SFTS in Macheng, Hubei Province [28]. It was used as the wild-type virus (SFTSV-WT) in this study, propagated in Vero cells, and stored as viral stocks at −80 °C until used.

### 2.2. 3′ and 5′ RACE Analyses of SFTSV-WT

To determine the 5′ and 3′ terminal sequences of the complementary viral RNAs (cRNAs), as described previously, 5′ and 3′ rapid amplification of cDNA ends (RACE) analyses were performed [29]. Briefly, for the 5′ RACE analysis, the viral RNA in the viral stocks was extracted using the MiniBEST Viral RNA/DNA Extraction Kit Ver. 5.0 (Takara, Shiga, Japan) and polyadenylated through the Poly (A) Tailing Kit (Thermo Fisher, Waltham, MA, USA), according to the manufacturer’s instructions. The polyadenylated RNA was then amplified by reverse transcription PCR (RT-PCR) using the PrimeScrpt^TM^ RT Reagent Kit with gDNA Eraser (Takara). Amplified products were purified and used for nucleotide sequence determination.

For the 3′ RACE analysis, the 3′ terminal sequence of the cRNA was obtained using a 5′ RACE Kit (BioTeke, PR6931, Beijing, China). The cDNA of the viral genome segments was synthesized by RT-PCR using segment-specific RT primers. A poly (C) tail was added to the 3′ end of the cDNAs using a terminal deoxynucleotide transferase (TdT). The cDNAs were then used as templates for PCR, and the amplified products were sequenced.

### 2.3. Construction of Plasmids for Reverse Genetics

Helper plasmids were constructed, as described previously [29]. Briefly, the RdRp and NP coding sequences obtained from viral RNA by RT-PCR were cloned into the pCAGGS vector using In-Fusion homologous recombinant technology (In-Fusion^®^ HD Cloning Kit, Clontech). The generated helper plasmids were labeled as pCAGGS-MC5-RdRp and pCAGGS-MC5-NP.

To construct the rescue plasmids, the full-length cDNAs of the viral genome L and M segments were obtained by fusing the above-corrected untranslated regions (UTR) sequences to the corresponding inner sequences via overlap PCR, and cloning them into the pT7 vector (kindly provided by Dr. Lei-Ke Zhang from the Wuhan Institute of Virology, Chinese Academy of Sciences) between a T7 promoter and a hepatitis delta virus ribozyme (HDVR) sequence in the viral complementary orientation using an In-Fusion^®^ HD Cloning Kit, resulting in pT7-MC5-L and pT7-MC5-M, respectively.

The modified S segment, with the coding region of the NSs (Δ2-282) replaced by the enhanced green fluorescent protein (eGFP), was obtained by overlap PCR. The fused segment was inserted into the T7 vector, as described above, resulting in the pT7-MC5-S-UTR-delNSs-eGFP plasmid. The clones were verified by sequencing.

### 2.4. Rescuing the Recombinant Virus

In addition, 1.5 × 10^5^ BSR-T7 cells were seeded into 12-well plates, cultured overnight at 37 °C, and transfected with pCAGGS-MC5-NP (500 ng), pCAGGS-MC5-RdRp (500 ng), pT7-MC5-L (1000 ng), pT7-MC5-M (1000 ng), and pT7-MC5-S-delNSs-eGFP (1000 ng) using Lipofectamine 3000 Transfection Reagent (Invitrogen), according to the manufacturer’s instructions. Four days post-transfection, virus-containing supernatants (passage 0, P0) were collected and used to infect fresh Vero cells. Four days after infection, the supernatants were collected (passage 1, P1), and the passaging was continued. Virus-containing supernatants from each passage were collected for the titration of viral infectivity and viral RNA copies.

### 2.5. Immunofluorescence Assay (IFA)

To visualize viral infection, infected cells were fixed with 4% formaldehyde and permeabilized with 0.2% Triton X-100. The expression of the NP and NSs proteins was detected using rabbit anti-NP and anti-NS antibodies [30]. AF 488/555-conjugated goat anti-rabbit IgG was added as a secondary antibody (Thermo Fisher Scientific), and the nuclei were stained with Hoechst 33258 (Beyotime Biotechnology, Shanghai, China). Fluorescence images were captured using a two-photon fluorescence microscope (A1RMP; Nikon, Tokyo, Japan).

### 2.6. Virus Titration by Immunostaining Assay

Viral infectivity was analyzed using an endpoint dilution assay. Although the reporter virus could be titrated using its own fluorescence, both viruses were titrated via immunostaining for a valid comparison with SFTSV-WT. Vero cells seeded in 96-well plates at 100% confluence were incubated with 10-fold serial dilutions of the viruses in DMEM supplemented with 2% FBS. Cells were incubated at 37 °C for 5 days before being fixed with 4% formaldehyde for IFA analysis, as described above, using an anti-NP antibody. Viral titers were calculated using the Reed-Muench method [31].

### 2.7. Quantitative Real-Time PCR (qRT-PCR)

Viral RNA in the supernatant was extracted using the MiniBEST Viral RNA/DNA Extraction Kit Ver. 5.0 (Takara), and the total RNA in the tissue was extracted using TRIzol (Promega, Madison, WI, USA), according to the supplier’s protocol. The NP of the S segment was amplified by RT-PCR with the primers NP-F: 5′-GCTGGCTCCGCGCATCTTCAC-3′ and NP-R: 5′-GGCACTCCAAGAGAAATATGG-3′, and cloned into the pMD-18T vector (Invitrogen, Waltham, MA, USA) and used as the plasmid standard. qRT-PCR was performed using the specific primers NP-F1: 5′-GCAGTTGGAATCAGGGA-3′ and NP-R1: 5′-CCCACTTGGACATGTGCT-3′, targeting the viral S segments and using Takara RR820A SYBR^®^ Premix Ex Taq™ II.

### 2.8. Viral Genome Sequencing

To determine the viral genome sequence, viral RNA was extracted from the supernatants of cells infected with SFTSV-WT or SFTSV-delNSs-eGFP. Viral RNA was converted to cDNA using the PrimeScrpt^TM^RT reagent kit with gDNA Eraser (Takara). cDNA was then prepared for sequencing following the standard protocol of the BGISEQ/MGISEQ NGS System (Beijing Genomics Institute, BGI, Beijing, China). The quality of the obtained data was verified using FastQC (version 0.11.9) and aligned to the SFTSV HBMC5 template sequence found in the NCBI (KY440769.1, KY440770.1, and KY440771.1) using Bowtie2 (version 2.3.4.3). Finally, the aligned sequencing results were analyzed using Tablet software (version 1.21.02.08).

### 2.9. Western Blotting

Vero cells infected with the virus at a multiplicity of infection (MOI) of 5 were lysed using RIPA Lysis Buffer (Beyotime Biotechnology) 48 h post-infection. Proteins were separated using SDS-PAGE and transferred onto a polyvinylidene difluoride membrane (Merck Millipore, Burlington, MA, USA). Western blotting was performed using primary antibodies against NP, NSs, or GAPDH [30]. The protein bands were detected using an enhanced chemiluminescence kit (Thermo Fisher Scientific).

### 2.10. Plaque Assay Using Immune-Staining

Vero cells cultured in 24-well plates at 100% confluence were infected with 10-fold serial dilutions of the viruses for 1 h, and the supernatant was replaced with fresh DMEM containing 2% FBS and 1% (*w*/*v*) Avicel (Millipore). The overlay was removed 5 days post-infection and fixed with 4% formaldehyde. The fixed cells were permeabilized with 0.2% Triton X-100 and probed with an anti-NP antibody and horseradish peroxidase (HRP)-labeled goat anti-rabbit IgG (Sigma, Tokyo, Japan). Cells were stained using a hydrogen peroxide (DAB) kit (Servicebio, Wuhan, China). Wells containing 10–100 immunofocus units (plaques) were selected for imaging and counting.

### 2.11. One-Step Growth Curve Analysis

Confluent monolayers of Vero and HeLa cells cultured in 24-well plates were infected with SFTSV-WT or SFTSV-delNSs-eGFP at an MOI of 5 at 37 °C for 1 h. The cells were then supplemented with a fresh medium containing 2% FBS. At 0, 12, 24, 48, and 72 h post-infection, supernatant samples were collected for infectivity analysis by titration, as described above.

### 2.12. Antiviral Drug Evaluation Assays

First, the cytotoxicity of T-705 (MedChemExpress, Monmouth Junction, NJ, USA) and CQ (Sigma) on Vero cells was tested using a cell counting kit-8 (CCK8; Beyotime Biotechnology), and the 50% cytotoxic concentration (CC_50_) was obtained, as described previously [32]. To evaluate the antiviral efficiency of the compounds, Vero cells (10^4^ cells/well) were cultured overnight in 96-well plates, pre-treated with different concentrations of T-705, CQ, or DMSO (as a control) for 1 h, and then infected with SFTSV-WT or SFTSV-delNSs-eGFP at an MOI of 0.1 for 1 h at 37 °C. Following this, the supernatant was replaced with fresh medium containing different concentrations of the corresponding compounds. At 72 h post-infection, viral RNA levels in the supernatant were determined using qRT-PCR, as described above. The 50% maximal effective concentration (EC_50_) and the selectivity index (SI = CC_50_/EC_50_) of each drug were calculated accordingly [32]. In addition, the cells infected with SFTSV-delNSs-eGFP were fixed by 4% paraformaldehyde for immunofluorescence counting using a high content system (HCS, PerkinElmer Operetta CLS™, Tokyo, Japan), and the EC_50_ was obtained, as previously described [33].

### 2.13. Mouse Experiments

To compare the pathogenicity of SFTSV-WT and SFTSV-delNSs-eGFP in vivo, 17–22-week-old female IFNAR^−/−^ C57/BL6J mice were used. The mice were infected intraperitoneally with 10 TCID_50_ (50% tissue culture infectious dose) of SFTSV-WT or SFTSV-delNSs-eGFP per mouse, and the control group was injected with PBS (*n* = 6/group). The mice were monitored daily for body weight, clinical symptoms, and survival rates for 9 days. The mice were euthanized when they exhibited two or more signs of severity, such as ruffled fur, hunched posture, lethargy, curling up, shaking, or a loss of more than 15% of body weight, or at the indicated predefined endpoints. Blood samples were collected for clinical chemistry analysis using a VetScan2 chemistry analyzer (Abaxis, Union City, CA, USA). The spleen was collected for viral load analysis by qRT-PCR and histopathological analysis.

### 2.14. Tissue Histology, Immunohistochemistry, and IFA

Spleen tissues from the infected mice were collected and fixed in 10% neutral formalin buffer. The tissues were processed for paraffin embedding and sectioning. Serial sections (3 µm-thick) were obtained and stained with hematoxylin and eosin (H&E), or processed for immunohistochemistry (IHC) or IFA. For IHC analysis, the sections were deparaffinized, and heat-mediated antigens were retrieved in citrate buffer (pH 6.0) (Sigma-Aldrich) using a microwave. After quenching endogenous peroxidase with 0.3% hydrogen peroxide, the sections were incubated with anti-NP antibody as the primary antibody and HRP-conjugated goat anti-rabbit IgG antibody as the secondary antibody, and stained using the RBD staining kit (Servicebio). For IFA, an AF 555-conjugated goat anti-rabbit IgG antibody was used as the secondary antibody.

### 2.15. Statistical Analyses

Data analysis was performed using Excel (Microsoft Corporation, Redmond, WA, USA) and GraphPad Prism 8.3.0 (538) software. Viral growth curves were analyzed using a two-way RM ANOVA. Statistical analysis of body weight loss was performed using a two-way ANOVA. A student’s *t*-test was used to compare differences in blood biochemical parameters and viral RNA copy numbers between the two groups. *p*-values of ≤0.05 were considered significant.

## 3. Results

### 3.1. Determination the SFTSV-HBMC5 Strain UTR Sequence

UTR sequences are crucial for reverse genetics. Therefore, we first determined the UTR sequences of SFTSV-HBMC5 by 3′ and 5′ RACE and compared them with the published SFTSV-HBMC5 sequences in GenBank. The results revealed that the UTRs of the L segment were identical to those in GenBank (KY440771.1); however, there were certain mutations in the UTRs of the M and S segments (Figure 1). For the M 3′ UTR, a U-to-G mutation was detected at position 128 in comparison to that in GenBank (KY440770.1). For the S segment, a pair of A-U bases was added at the 10th position, and a U-to-C mutation was detected at position 22 in the 3′ UTR, compared to the GenBank sequence (KY440769.1). All corrected SFTSV-HBMC5 UTRs showed typical bunyavirus UTR characters that the first 9 nt of the 5′ and 3′ UTRs were complementary and relatively conserved among all segments (Figure 1, sequences in the blue background). After these 9 bp, the UTRs sequence was segment-specific and still exhibited complementarity (Figure 1, sequences in the grey background). RACE-corrected UTR sequences were used to construct rescue plasmids.

### 3.2. Recovery of Recombinant SFTSV-delNSs-eGFP

After constructing and verifying the helper and rescue plasmids, we recovered the infectious SFTSV-delNS-eGFP virus by co-transfecting the helper plasmids pCAGGS-MC5-NP and pCAGGS-MC5-RdRp with the genome plasmids pT7-MC5-L, pT7-MC5-M, and pT7-MC5-S-delNSs-eGFP into BSR-T7 cells. The transfected supernatant was continuously transferred to Vero cells, and the fluorescence of the infected cells in different generations was observed (Figure 2A). As shown in Figure 2B, only a minimal fluorescence signal was initially detected in passage 1 (P1) cells, and the number of infected cells increased with successive passaging. At P5, a large number of dispersed infected cells was observed, and almost all cells were infected at P7. The viral titer and viral RNA levels in the supernatant were also measured during passaging. The results revealed that the repeated passaging of SFTSV-delNSs-eGFP substantially increased virus yield from 4.88 × 10^3^ TCID_50_/mL in P3 to 1.68 × 10^7^ TCID_50_/mL in P8, and raised viral RNA from 4.5 × 10^7^ copies/mL in P3 to 2.2 × 10^10^ copies/mL in P8 (Figure 2C). These results indicate that infectious SFTSV-delNSs-eGFP was successfully rescued.

To identify any mutations in the generated virus, the complete genomes of SFTSV-delNSs-eGFP and its parent virus, SFTSV-WT, were sequenced and compared with the SFTSV-HBMC5 sequences in GenBank. In addition to the differences in the UTR shown in Figure 1 (all present in SFTSV-delNSs and its parent virus, as expected), four nucleotide changes were found in the coding region (Table 1). Two mutations were identified in the coding region of the L segment. The first was an A-to-G mutation at 1199 nt that results in an amino acid change from Asn to Asp at the 395 residue of the RdRp protein. The second was a C-to-T mutation at 4501 nt, which was synonymous at the protein level. In the coding region of the M segment, two mutations were detected: a C-to-T change at 403 nt and an A-to-G change at 3132 nt, resulting in a Leu to Phe mutation in Gn and a synonymous mutation in Gc, respectively. All mutations were also present in the parent viruses, indicating that these mutations occurred naturally during the adaptation of the parental virus to cell culture. For the S segment, except for the designed changes, the coding sequences of SFTSV-delNSs-eGFP and its parental virus were consistent with the sequences in GenBank.

### 3.3. Characterization of SFTSV-delNSs-eGFP In Vitro

Since SFTSV-delNSs-eGFP was constructed by replacing NSs with eGFP, we first checked NSs protein expression in virus-infected cells. Vero cells infected with SFTSV-delNSs-eGFP or SFTSV-WT were collected to detect NP and NSs protein expression by western blotting. As shown in Figure 3A, similar NP protein levels were detected in the cells infected with both viruses, but NSs were detected only in SFTSV-WT-infected cells and not in SFTSV-delNSs-eGFP-infected cells. GAPDH was used as a loading control. We also investigated the formation of NS inclusion bodies in infected cells by immunofluorescence staining. As shown in Figure 3B, numerous NSs (red) and inclusion bodies were observed in SFTSV-WT-infected cells. In contrast, no red fluorescence for NSs was detected in SFTSV-delNSs-eGFP-infected cells, and green fluorescence from eGFP was observed instead, suggesting the replacement of NSs with eGFP.

To analyze the infective properties of SFTSV-delNSs-eGFP, an immunostaining assay was performed on the virus-containing supernatant collected from P8 (Figure 3C). Rescued SFTSV-delNSs-eGFP presented foci with a size and morphology similar to those of SFTSV-WT. Furthermore, the growth curves of SFTSV-WT and SFTSV-delNSs-eGFP were compared in Vero (IFN-incompetent) and HeLa (IFN-competent) cells at an MOI of 5. As shown in Figure 3D, the parental virus SFTSV-WT and recombinant SFTSV-delNSs-eGFP both replicated efficiently in Vero cells with similar growth kinetics (*p* > 0.05), achieving peak titers of 5.63 × 10^7^ and 2.28 × 10^7^ TCID_50_/mL at 24 h post-infection, respectively (Figure 3D, left). However, in HeLa cells, SFTSV-WT replicated more efficiently than SFTSV-delNS-eGFP. The virus yield of SFTSV-WT in the supernatant gradually increased with the increase in infection time and reached a peak titer of 2.14 × 10^7^ TCID_50_/mL at 72 h post-infection. For SFTSV-delNSs-eGFP, the titer achieved a peak of 3.58 × 10^5^ TCID_50_/mL at 24 h post-infection and then dropped slightly. Statistical analysis showed that the growth kinetics of SFTSV-delNSs-eGFP were significantly slower than those of SFTSV-WT in HeLa cells (*p* < 0.0001) (Figure 3D, right).

### 3.4. SFTSV-delNSs-eGFP Could Be Used as a Reporter Virus for the Evaluation of Antiviral Drugs

Since SFTSV-delNSs-eGFP could be visualized through eGFP fluorescence, we further investigated whether fluorescent SFTSV could be used for antiviral drug screening. To this end, two drugs that have reported antiviral properties against SFTSV, T-705 and CQ, were selected for testing. The cytotoxicity of the two compounds in Vero cells was measured using a standard CCK8 assay, and their antiviral activities were tested in SFTSV-infected cells (0.1 MOI) treated with these compounds or DMSO as a control. The dose-response curves were determined by the quantification of viral RNA copy numbers through qRT-PCR in the supernatant (Figure 4A,B,D,E) and visualization of fluorescence through an HCS of the infected cells (Figure 4C,F) at 72 h post-infection. As demonstrated in Figure 4, the EC_50_ of T-705 against SFTSV-WT (Figure 4A) and SFTSV-delNSs-eGFP (Figure 4B) measured by qPCR were 32.0 μM and 12.4 μM, respectively, and the EC_50_ determined through HCS was 20.9 μM (Figure 4C). In addition, the EC_50_ of CQ against SFTSV-WT and SFTSV-delNSs-eGFP were measured as 10.1 μM (Figure 4D) and 4.5 μM (Figure 4E), respectively, by RT-qPCR, and the EC_50_ of CQ against SFTSV-delNSs-eGFP calculated through fluorescence was 15.1 μM (Figure 4F). These results showed that the inhibitory efficacy of T-705 and CQ determined by HCS was in a similar range to that obtained by qRT-PCR using either SFTSV-WT or SFTSV-delNSs-eGFP, indicating that SFTSV-delNSs-eGFP is suitable as a reporter virus for the high-throughput screening of anti-SFTSV drugs.

### 3.5. The Attenuated Pathogenicity of SFTSV-delNSs-eGFP In Vivo

Because NSs are virulence factors for SFTSV, NSs deletion was predicted to reduce the pathogenicity of SFTSV. To compare the pathogenicity of SFTSV-delNSs-eGFP and its parental virus, IFNAR^−/−^ C57BL/6J mice were infected with 10 TCID_50_/mouse of either SFTSV-WT or SFTSV-delNSs-eGFP. The body weights and clinical signs of the infected mice were measured. Five of the six SFTSV-WT-infected mice reached a clinically defined endpoint by day four post-infection, whereas all SFTSV-delNSs-eGFP-infected mice survived infection (Figure 5A). All IFNAR^−/−^ mice infected with SFTSV-WT developed clinical signs of disease, including severe weight loss (Figure 5B), ruffled fur, a hunched posture, and lethargy. Biochemical analysis of the blood samples showed significantly elevated levels of alanine aminotransferase (ALT), aspartate aminotransferase (AST), and lactate dehydrogenase (LDH) (Figure 5C). However, IFNAR^−/−^ mice infected with SFTSV-delNSs-eGFP did not show any obvious symptoms of illness and showed a stable body weight (Figure 5B) and ALT/AST/LDH levels (Figure 5C). Viral RNA was detected using qRT-PCR in the spleen, which is one of the main target tissues for SFTSV infection [34,35]. As shown in Figure 5D, the copy number of viral RNA in the SFTSV-WT-infected spleen was significantly higher than that in the SFTSV-delNSs-eGFP-infected spleen (*p* < 0.01), whereas the latter had no significant difference from that of the uninfected mice. H&E staining showed that, compared to spleens from PBS-infected mice, SFTSV-WT infection resulted in the prominent disruption of the white pulp of the spleen; however, SFTSV-delNSs-eGFP infection resulted in the minimal disruption of tissue architecture (Figure 5E, upper panel). Immunohistochemical and immunofluorescence staining for SFTSV NP also confirmed the presence of viral infection in the SFTSV-WT-infected spleen, but not in the spleen infected with SFTSV-delNSs-eGFP (Figure 5E, middle and lower panels). These results showed that SFTSV-delNSs-eGFP attenuated pathogenicity compared to the parental WT virus in vivo.

## 4. Discussion

SFTS is an emerging infectious disease with substantial fatalities and no specific medicines or vaccines. The construction of a fluorescent reporter virus with improved biosafety can facilitate the high-throughput screening of antiviral drugs. We previously constructed a reverse genetic system based on the SFTSV-WCH strain, but the rescued virus had a low replication capacity with a titer of only 7.9 × 10^5^ PFU/mL [29]. In the present study, SFTSV-HBMC5 was selected as the parent virus to construct a fluorescent reporter virus. SFTSV-HBMC5 was isolated from a patient with SFTS in Macheng and effectively replicated in Vero cells with a titer of 1 × 10^8^ TCID_50_/mL [28]. Using this virus as the backbone, we successfully constructed a reporter virus, SFTSV-delNSs-eGFP.

NSs are important virulence factors of SFTSV, and SFTSV-delNSs-eGFP was constructed by replacing the NSs coding region (Δ2-282) with eGFP, according to the strategy by Brennan et al. [36]. Western blotting and IFA confirmed that NSs were successfully eliminated and eGFP was expressed in the virus. Consistent with the study by Brenann et al. [36], the growth properties of SFTSV-delNSs-eGFP and SFTSV-WT were not significantly different in Vero cells, which are type I IFN-incompetent [37]. As expected, the growth of SFTSV-delNSs-eGFP in HeLa cells, which express type I IFN, was drastically reduced compared to that of SFTSV-WT (Figure 3), confirming that NSs are key factors in viral replication in interferon-competent cells [37].

Benefitting from its fluorescence, SFTSV-delNS-eGFP has the potential for use in high-throughput drug screening. In this study, we tested the antiviral efficacy of T-705 and CQ using the SFTSV-delNSs-eGFP reporter and compared it with that of the wild-type virus (Figure 4). Previously, the EC_50_ of T-705 against SFTSV-WT was reported to range from 6 μM to 25 μM in Vero cells [23,38]. Our results showed that the EC_50_ of T-705 against SFTSV-WT and SFTSV-delNSs-eGFP was 32.0 μM and 12.4 μM, respectively, when measured by qRT-PCR, whereas HCS determined the EC_50_ of T-705 against SFTSV-delNSs-eGFP as 20.9 μM. Although our results showed variations between the different methods, they are largely in agreement with previous reports. Similarly, the EC_50_ of CQ against SFTSV was measured at 15.1 μM by HCS, which was similar to the EC_50_ of 10.1 μM measured in SFTSV-WT using the traditional qRT-PCR method. As HCS can greatly simplify the process and shorten the experimental period, SFTSV-delNSs-eGFP is a potent tool for the high-throughput antiviral drug screening of SFTSV. Although Brennan et al. constructed a fluorescent reporter SFTSV based on another strain (HB29), they did not evaluate its potential for high-throughput drug screening [36]. Here, we confirmed the potent potential of SFTSV-delNSs-eGFP for high-throughput drug screening, as well as the potential of their virus. Notably, NSs were missing from the reported virus, and the efficacy obtained from the reported virus was not identical to that of the WT virus. Therefore, we recommend using the reporter virus for the high-throughput screening of antiviral candidates and using the WT virus to further confirm the identified candidates.

Our results demonstrated that SFTSV-delNSs-eGFP lost infectivity when inoculated at 10 TCID_50_/mouse into the IFNAR^−/−^ C57BL/6J mice, whereas its parental virus was lethal at the same dose. IFNAR^−/−^ C57BL/6J mice exhibit type I IFN responses that are incompetent when the IFN receptor is knocked out. The lack of SFTSV-delNS-eGFP infectivity in IFNAR^−/−^ C57BL/6J mice suggests that NSs may play important roles other than those involved in type I IFN induction and signaling. Recently, Bryden et al. revealed that delNS SFTSV can induce a protective adaptive immune response and type II IFN (IFN-γ) secretion, which play a key role in protecting type I IFN-deficient mice from delNSs SFTSV infection [19]. The attenuated pathogenicity of SFTSV-delNSs-eGFP indicates improved laboratory biosafety for future applications.

In conclusion, SFTSV-delNSs-eGFP is a reporter virus with efficient replication properties and better laboratory biosafety than SFTSV-WT, and can potentially be used for high-throughput antiviral drug screening and virology studies in the future.

## Figures and Tables

**Figure 1 viruses-15-01147-f001:**
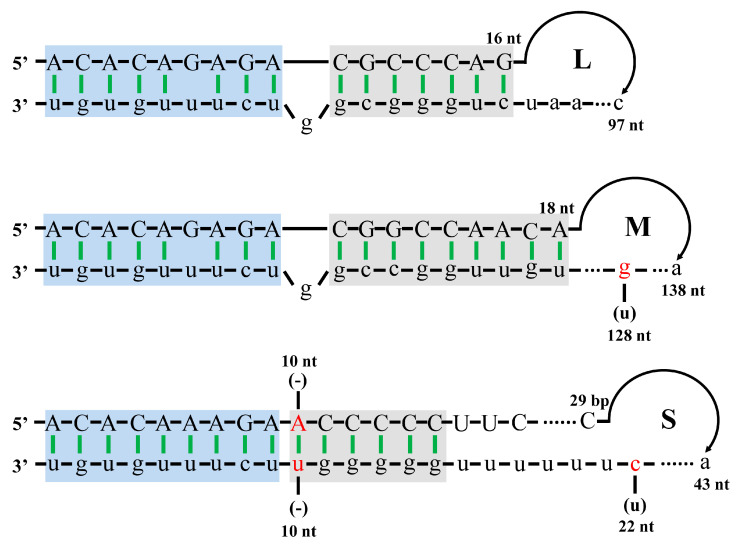
The 3′ and 5′ UTRs of complementary viral RNA (cRNA) of SFTSV-WT. The comparison results between the sequences detected by 3’ and 5’ RACE and those of SFTSV-HBMC5 published in GenBank are shown. The 5′ UTR sequences are shown in capital letters, and the 3′ UTR sequences are shown in lowercase letters. The red bases indicate sequences corrected by 3’ and 5’ RACE analyses that differ from those published in GenBank (in parentheses); (-) indicates that the sequence published by GenBank is missing in the corresponding position. The blue box represents the relatively conserved region among the L, M, and S segments, while the grey box represents the specific complementary region of each segment. The base digits at the ends of the 5’ and 3’ UTR correspond to the length of the UTR.

**Figure 2 viruses-15-01147-f002:**
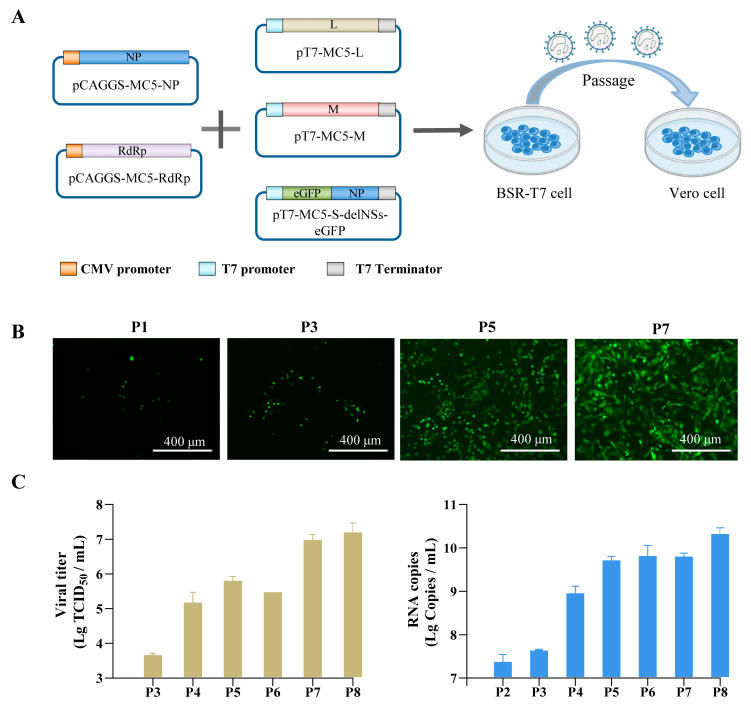
Rescue of SFTSV-delNSs-eGFP. (**A**) The flow chart of SFTSV-delNSs-eGFP generation. The infectious virus was obtained by co-transfecting the helper plasmids pCAGGS-MC5-NP and pCAGGS-MC5-RdRp with rescue plasmids pT7-MC5-L, pT7-MC5-M, and pT7-MC5-S-delNSs-eGFP in BSR-T7 cells and continuing passaging of the supernatants in Vero cells. (**B**) The fluorescence of the infected cells in different passages (P1, P3, P5, and P7) was observed. Scale bar = 400 μm. (**C**) Viral titers and viral RNA copy levels of the collected supernatants from P2 to P8.

**Figure 3 viruses-15-01147-f003:**
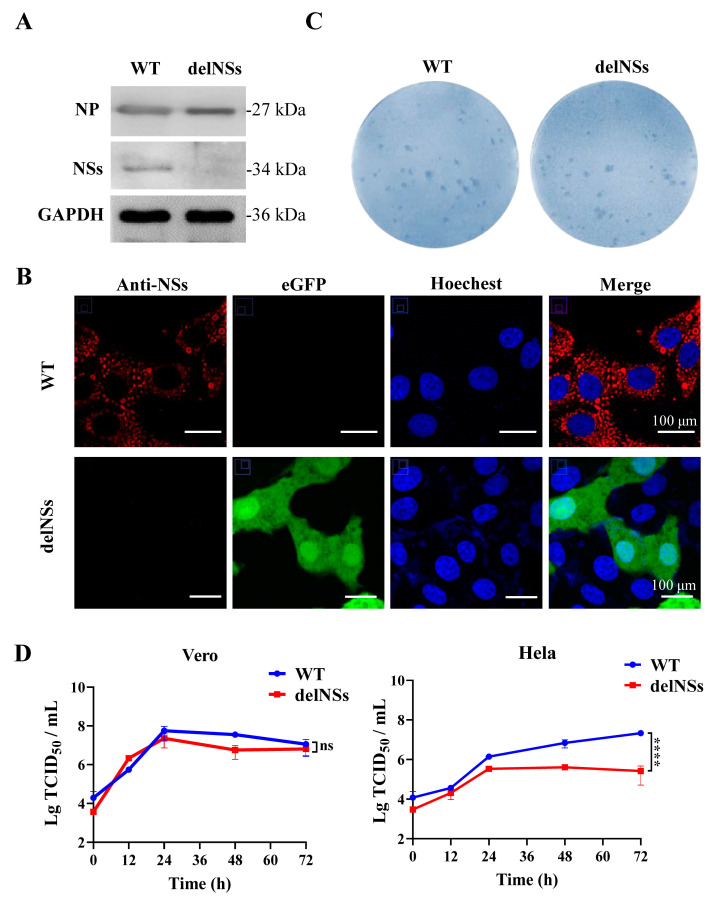
Characterization of SFTSV-delNSs-eGFP in vitro. (**A**) Western blot analysis of nucleoprotein (NP) and nonstructural protein (NSs) expression in infected cells. Vero cells were infected with SFTSV-delNSs-eGFP or SFTSV-WT at an MOI of 5, and cellular samples were harvested for western blotting using an anti-NP, anti-NSs, or anti-GAPDH antibody. (**B**) Fluorescence confocal observation of NSs inclusion bodies. Vero cells infected with SFTSV-delNSs-eGFP or SFTSV-WT at an MOI of 1 were fixed at 24 h post-infection and stained for immunofluorescence assay by an anti-NS_S_ antibody. Scale bar = 100 μm. (**C**) Comparison of immune-stained foci of SFTSV-delNSs-eGFP or SFTSV-WT. (**D**) One-step growth curves of SFTSV-delNSs-eGFP or SFTSV-WT conducted in Vero and HeLa cells at an MOI of 5. The two-way RM ANOVA analysis was used for statistical analysis of the overall curve; ns indicates no significant difference (*p* > 0.05) and **** indicates *p* < 0.0001. WT, SFTSV-WT; delNSs, SFTSV-delNSs-eGFP.

**Figure 4 viruses-15-01147-f004:**
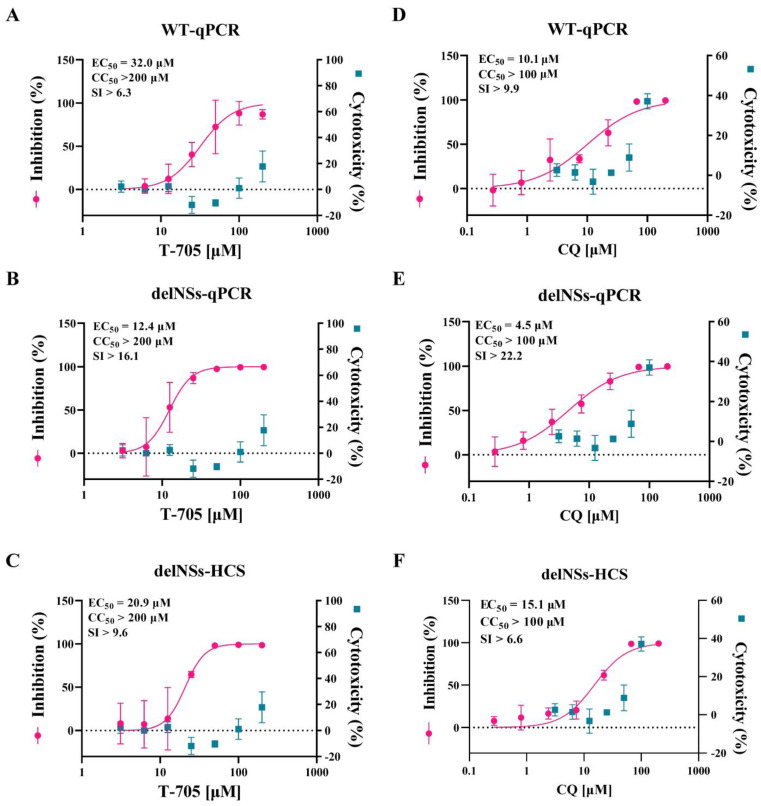
Antiviral efficacy of T−705 (**A**–**C**) and CQ (**D**–**F**) measured with SFTSV-delNSs-eGFP and SFTSV-WT. (**A**,**B**,**D**,**E)** showed the efficacy measured by qRT-PCR (qPCR) of both viruses, while (**C**,**F**) showed efficacy measured by high content screening (HCS) of fluorescence using SFTSV-delNSs-eGFP. Vero cells were infected with SFTSV-WT or SFTSV-delNSs-eGFP at an MOI of 0.1 and treated with different doses of the indicated antivirals for 72 h. The viral yield in the cell supernatant was then quantified by qRT-PCR, or the infected cells were quantified using HCS. The cytotoxicity of these drugs was measured by a CCK-8 assay. The left and right Y-axes of the graphs represent the mean % inhibition of the virus yield and the cytotoxicity of the drugs, respectively. CC_50_, the 50% cytotoxic concentration; EC_50_, the 50% maximal effective concentration; SI, the selectivity index (CC_50_/EC_50_).

**Figure 5 viruses-15-01147-f005:**
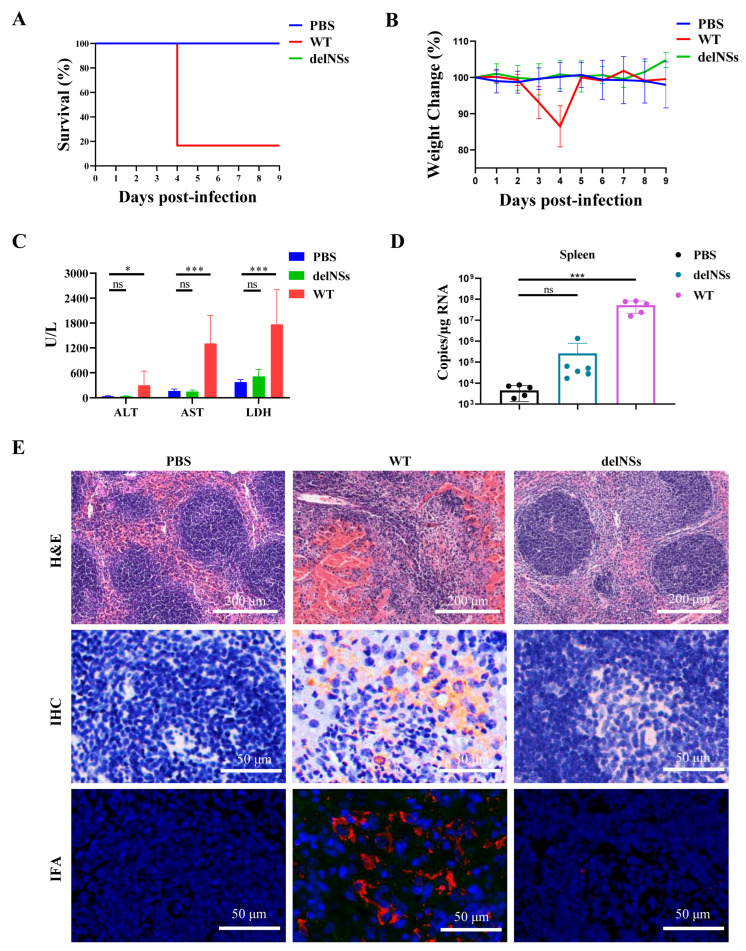
Comparison of the pathogenicity of SFTSV-delNSs-eGFP and SFTSV-WT in vivo. C57BL/6J IFNAR^−/−^ mice were infected with 10 TCID_50_ of SFTSV-delNSs-eGFP or SFTSV-WT, or mock infected with PBS (*n* = 6 per group). (**A**) The survival curves. The infected mice were euthanatized until they reached the humane endpoints, or 9 days post-infection. (**B**) The body weight loss curves. Notably, four days post-infection, the body weight curve of the WT group was derived from only one surviving mouse. (**C**) Blood biochemical parameters, including alanine aminotransferase (ALT), aspartate aminotransferase (AST), and lactate dehydrogenase (LDH). (**D**) Viral RNA levels in the spleen of the infected mice. (**E**) Histopathological analyses of the spleens in infected mice. Hematoxylin and eosin (H&E) staining (upper panel), IHC (middle panel), and IFA analyses (lower panel) using an anti-NP antibody were performed on spleens. A one-way ANOVA was used for statistical analysis; ns indicates no significant difference (*p* > 0.05); * *p* < 0.01; *** *p* < 0.001.

**Table 1 viruses-15-01147-t001:** Comparison of the inner sequences of SFTSV-WT and SFTSV-delNSs-eGFP and GenBank sequences of SFTSV-HBMC5.

Segments	Position		GenBank	SFTSV- delNSs-eGFP	SFTSV- WT
	nt	Protein (aa)			
L	1199	RdRp (395)	A (Asn)	G (Asp)	G (Asp)
	4501	RdRp (1495)	C (Ile)	T (Ile)	T (Ile)
M	403	Gn (109)	C (Leu)	T (Phe)	T (Phe)
	3132	Gc (479)	A (Leu)	G (Leu)	G (Leu)

## Data Availability

The data that support the findings of this study are available from the corresponding author upon reasonable request.

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
