# Peer review of "Construction and Characterization of Severe Fever with Thrombocytopenia Syndrome Virus with a Fluorescent Reporter for Antiviral Drug Screening"

_viruses, 2023, doi:10.3390/v15051147_

Round 1
Reviewer 1 Report
Comments on Wang et al., “Construction and characterization of a fluorescent reporter for SFTSV for antiviral drug screening”
This paper describes the construction and testing of a recombinant SFTSV virus that lacks its canonical virulence factor, NSs, and replaces it with the gene for green fluorescent protein. The work is carefully described and follows very closely to the approach and methods used to construct the homologous recombinant viral mutant of RVFV (another phenuivirus from the genus phlebovirus) as described by Brennan and coworkers in 2017.
The virus was constructed by reverse genetics, which is notoriously fickle, and the authors did a good job of describing their efforts to isolate, amplify, and adapt the virus to growth in cell culture.
As expected, the lack of NSs results in an attenuated virus, with demonstrated defect in suppressing the cellular interferon response to infection. This makes this new mutant virus safer to work with in the laboratory and, as the authors show, will make it useful for cell-based drug screening programs.
Overall the paper is very well written, I only have minor comments to address:
1) The predicted secondary structures for the panhandles of the L, M, and S segments of the cRNA in figure 1 look as expected for a typical bunyavirus except for the S segment, since the image shows no bulge after 9 nucleotides from the end. Studies have suggested that the bulge is an essential feature, and may inhibit recognition of the viral RNA by the PKR response. Because shifting the top strand to the right by one or even two or three nts would result in a bulge and would still provide as many base pairs (by inclusion of one or two G:U pairs and possibly a new G:C base pair at the proximal end of the ‘gray’ helix). What are the nucleotides on the top strand just following the last “C”? is it possible that they are A’s or G’s that could provide the base pairing energy to form the bulged helix? An effort should be made to see if a bulged helix can be modeled.
2) Although not strictly necessary for this paper, efforts should be made to characterize the individual effects of the spontaneous mutations that arose during adaptation.
3) The antiviral drug favipiravir is misspelled in the abstract and on line 350.
4) In the Discussion section, there are noun-verb agreement problems when speaking about the protein (or gene) NSs. This is not a plural word, even though it ends with “s”. Therefore, the verb form should match with a singular noun.
Overall, English is sound- there are a few places that need some attention as noted above.
Reviewer 2 Report
Wang et al. reported about developing recombinant SFTSV containing fluorescent reporter, which is useful for performing HTS to identify novel drug candidates.
Overall, the manuscript has been described well. However, it needs some additional explanation to improve the current manuscript as below.
1. The title does not support the current work adequately. Reviewer propose the title as “Construction and Characterization of Severe Fever with Thrombocytopenia Syndrome Virus with a fluorescent reporter for Antiviral Drug Screening.
2. Throughout the manuscript, the term “in vitro” should be described in italic font.
3. In line 62, the original works should be referred rather than referring the review to appreciate the original articles.
4. For the virus titration by immunostaining assay, reviewer wonder why the incubation period was 5 days? Five days incubation would allow the virus propagation for the second, third, or more replication cycles, which should lead to the mis-counting the original infectious particle titer as long as using IFA.
5. In line 216, 10-50%? rather than 10 50%.
6. In table 1, why only the nt 1199 in L was described in bold font with underline?
7. Figure 3C, clearer picture would be requested.
8. Some parts in the result section, methodology was described. Description of the methodology should be minimized in the result section.
9. In figure 5D, why the authors only measure the virus copy number from the spleen? How about the other organs and serum?
10. For example, the lines 422 and 425 as well as Figure 2C, why the authors use two different units for the virus titration? It might be impossible to simply compare them with the different units.
11. In line 434, Fig.2 must be Fig.3.
12. Authors should emphasize the strong point for the development of the recombinant SFTSV HBMC5 strain with a fluorescent reporter, since similar reporter viruses have been already reported from other groups.
No problem.
